# Beyond personal factors: Multilevel determinants of childhood stunting in Indonesia

**Tri Mulyaningsih**[1]*, **Itismita Mohanty**[2], **Vitri Widyaningsih**[3], **Tesfaye Alemayehu Gebremedhin**[4], **Riyana Miranti**[4], **Vincent Hadi Wiyono**[1]

**1** Department of Economics, Faculty of Economics and Business, Universitas Sebelas Maret, Surakarta, Central Java, Indonesia, **2** Health Research Institute, University of Canberra, Canberra, Australia, **3** Faculty of Medicine, Universitas Sebelas Maret, Surakarta, Central Java, Indonesia, **4** Canberra School of Politics, Economics and Society, Faculty of Business, Government and Law, University of Canberra, Canberra, Australia

* trimulyaningsih@staff.uns.ac.id

## Abstract

### Background

Stunting is still a major public health problem in low- and middle-income countries, including Indonesia. Previous studies have reported the complexities associated with understanding the determinants of stunting. This study aimed to examine the household-, subdistrict- and province-level determinants of stunting in Indonesia using a multilevel hierarchical mixed effects model.

### Methods

We analyzed data for 8045 children taken from the 2007 and 2014 waves of the Indonesian Family and Life Surveys (IFLS). We included individual-, family-/household- and community-level variables in the analyses. A multilevel mixed effects model was employed to take into account the hierarchical structure of the data. Moreover, the model captured the effect of unobserved household-, subdistrict- and province-level characteristics on the probability of children being stunted.

### Results

Our findings showed that the odds of childhood stunting vary significantly not only by individual child- and household-level characteristics but also by province- and subdistrict-level characteristics. Among the child-level covariates included in our model, dietary habits, neonatal weight, a history of infection, and sex significantly affected the risk of stunting. Household wealth status and parental education are significant household-level covariates associated with a higher risk of stunting. Finally, the risk of stunting is higher for children living in communities without access to water, sanitation and hygiene.

**Data Availability Statement:** The data are available in a public repository from the https://www.rand.org/well-being/social-and-behavioral-policy/data/FLS/IFLS/download.html.

**Funding:** Authors received the award: Tri Mulyaningsih, Vitri Widyaningsih and Vincent Hadi Wiyono Funder: Universitas Sebelas Maret Grant Contract number 452/UN27.21/PN/2020 URL: www.uns.ac.id The funders had no role in study design, data collection and analysis, decision to publish, or preparation of the manuscript.

**Competing interests:** The authors have declared that no competing interests exist.

## Conclusions

Stunting is associated with not only child-level characteristics but also family- and community-level characteristics. Hence, interventions to reduce stunting should also take into account family and community characteristics to achieve effective outcomes.

## 1. Introduction

Stunting is an ongoing issue in many low- and middle-income countries. UNICEF/WHO and the World Bank [1] indicate that the number of stunted children is approximately 151 million, accounting for 22.2% of the children in the world. Moreover, the proportion of stunted children is concentrated in low-income (16%) and lower-middle-income (47%) countries compared to upper-middle-income (27%) and high-income (10%) countries [1]. Approximately 83.8 million stunted children live in Asia, mainly in Southern and southeastern Asia, 58.7 million in Africa and 5.1 million in Latin America and the Caribbean.

Indonesia is one of the countries with a high burden of malnutrition, including stunting [1]. Child health outcomes are poor, even though the Indonesian economy is the largest in Southeast Asia and the 17th largest in the world [2]. Data published by the Ministry of Health show that the incidence of stunting among children aged five years and below remains high at 30.8% [3]. The World Bank (2020) [4] noted that Indonesia has underperformed in terms of reducing the level of stunting compared to other upper-middle-income countries and other countries in the region. Given the high prevalence of stunting and its impact on children's cognitive development, the productivity level of Indonesia's next generation is predicted to be half of its potential [4]. Therefore, tackling child stunting remains a major government commitment, as asserted in the Indonesia Medium Development Goals 2015–2019 and 2020–2024 [5, 6].

The wider literature on stunting reveals that various child-, parental-, household- and community-level characteristics are associated with stunting [7–13]. At the parental and household levels, several dietary and socioeconomic factors have been shown to be correlated with the risk of stunting. With regard to risk factors, Beal et al. [7], for example, established that the risk of stunting in Indonesia is higher in households that have no access to safe drinking water. Household wealth status is another significant predictor, as children coming from poor households are more likely to be stunted [11, 12]. Meanwhile, at the community level, the prevalence of stunting has been shown to be higher in communities that lack access to health care [7]. In terms of protective factors for stunting, previous studies have shown that the likelihood of stunting is lower in communities where antenatal care services and integrated health and nutrition services are available [8, 9, 13]. In addition, consumption of diverse food within the household has also been found to lower the likelihood of stunting [14]. Furthermore, parental education has been shown to be significant, with children raised by educated parents having a lower risk of being stunted. Semba et al. [11] explored the channels through which parental education impacts stunting and argued that educated parents provide more care (in the form of having their children immunized and providing them with vitamin A and iodized salt), which would in turn lower the risk of stunting.

The prevalence of stunting in Indonesia varies by region, as illustrated in Fig 1 below [3]. It improved across all provinces between 2013 and 2018, except in East Kalimantan (Kalimantan Timur). The capital city of Jakarta Province had the lowest prevalence in 2018 at 17.7%, whereas East Nusa Tenggara recorded the highest at 42.6%. Provinces in the eastern part of Indonesia, where many development indicators lag behind other regions, have a higher

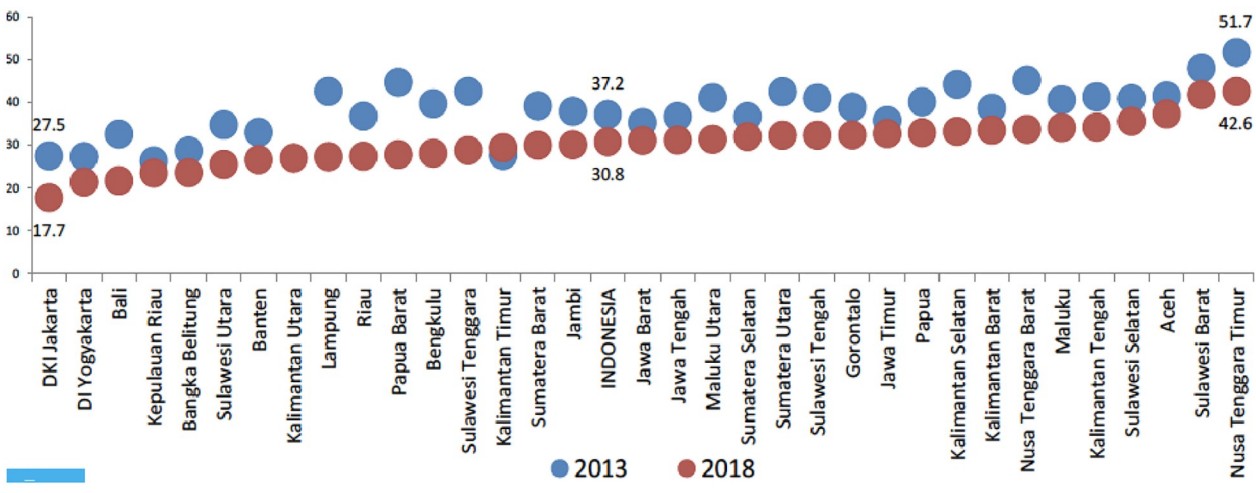

**Fig 1. Stunting prevalence across provinces in Indonesia.** Source: Basic Health Survey (2018).

prevalence of stunting. The World Bank (2020) [4] has also highlighted the regional variation in the incidence of stunting in Indonesia and further noted that the risk of stunting was higher in poor and populous districts where access to basic infrastructure of water, sanitation and hygiene (WASH) was lacking.

This paper examines the individual-, household-, subdistrict- and provincial-level determinants of stunting using a multilevel mixed effects model. Methodologically, this is an improvement over most of the previous literature in the field [see, for example, 13–15], as earlier studies examined the determinants of stunting in Indonesia using logistic and probit regressions. A multilevel model was appropriate for our analysis because of the hierarchical or clustered structure of our data. Young children living in the same household and community can be expected to have more similar stunting risks compared to those living in different households and communities [11, 12, 15]. Furthermore, children in the same province are more likely to have the same risk of being stunted because they have similar access to health care services and other infrastructure [8, 9, 13]. In addition, parental characteristics and household socioeconomic background have an influence on children's eating habits and nutritional status [14, 15]. Multilevel modeling enables us to investigate individual heterogeneities and the heterogeneities between clusters and improves the estimation techniques used in previous stunting studies [see, for example, 13–15]. In addition, taking into account the clustering in the data generates more reliable standard errors of regression coefficients [16].

This paper addresses two main research questions: Do variations at the province, subdistrict, household and individual levels explain childhood stunting in Indonesia? What are the multilevel determinants of childhood stunting in Indonesia? The remaining parts of the paper are organized as follows. Section 2 discusses the data and methodology used in the paper, including the outcome variables and various control variables included in the estimation. Section 3 presents and discusses the results of our estimation. The last section presents the conclusions and provides policy recommendations based on the findings of the study.

## 2. Data and methods

### 2.1. Data source

An open access and rich dataset from the Indonesian Family Life Survey (IFLS) was used to estimate the multilevel determinants of stunting across households, subdistricts and

provinces in Indonesia (https://www.rand.org/well-being/social-and-behavioral-policy/data/FLS/IFLS.html). The IFLS is a longitudinal survey representing 83% of the Indonesian population in 1993. The survey has five waves of data collected in 1993, 1997, 2000, 2007–2008 and 2014–2015. The survey was conducted by the RAND Institute with the cooperation of local universities and research centers in Indonesia. The first wave of the survey covered only 13 provinces, but the number was broadened to include more than 20 provinces in the last round to capture respondents' mobility to other provinces. Ethical clearances for the surveys were provided by institutional review boards (IRBs) in the United States and Gadjah Mada University (UGM) for IFLS waves 3, 4 and 5 and by the University of Indonesia for IFLS waves 1 and 2.

This particular study focuses on the last two waves of IFLS—wave 4 (conducted between 2007 and 2008) and wave 5 (conducted between 2014 and 2015)–analyzed as repeated cross-sectional surveys. There were 13,500 households and 43,000 individuals interviewed in the fourth wave of the study. The number of respondents increased in the fifth wave to 15,900 households and more than 50,000 individuals. This study only covers young children aged five years old and below. The dataset includes a total of 8,290 young children, with 4,142 in wave 4 and 4,148 in wave 5. Using child anthropometry information, we excluded children identified as having biologically implausible values (n = 185, 2.2%) based on z-scores of height, following the World Health Organization (WHO) child growth standard, which reduced the number of children to 8,105. In addition, we removed young children for whom complete information on household and community characteristics was not available (n = 60, 0.7%). Hence, the number of young children included in our final analysis was 8,045. The proportion of respondents with missing values was insignificant (3%).

Four hierarchical levels were considered in our analysis: individual (child), household, sub-district and province. Children (the lowest level in our mixed effects hierarchical model) were nested within households (level two). There were 6,437 households with 8,045 children aged 5 and below in the two waves of the IFLS. Approximately 20% of the total households had more than one child aged 5 and below. Households were nested within subdistricts (*Kecamatan* in Indonesian). There were 1,332 subdistricts recorded in the two waves of the survey. We considered the subdistrict as the third level in our model, as service provision and quality of care in primary health centers (PHCs) or *puskesmas* (which are expected to impact childhood stunting) are determined at the subdistrict level. The fourth level was the province, within which subdistricts were nested. We considered provinces as the highest level in our analysis; there were 21 provinces in the two waves of the IFLS used in our study.

The IFLS provides a wide range of information, including data on birth history, anthropometry of children (i.e., height, weight and body mass index), clinical and subclinical infection, dietary habits and demographic characteristics. Child health status data include both a self-reported measure of general health status and a biomarker measurement conducted by a nurse. Furthermore, the household data consist of information on parental education and parental anthropometry, antenatal care during pregnancy, consumption and wealth, and household access to WASH. Finally, the IFLS also includes other community-level data, consisting of access to nutrition-specific intervention programs in the PHCs in children's neighborhoods.

## 2.2. Outcome variable

According to the 2005 World Health Organization child growth standard and Indonesian Ministry of Health guidelines [17], stunting is a measure of children's nutritional status based on their height. Nutritional status is determined by comparing the height of children with

their peers of the same age. Height was measured during the survey by trained interviewers. Then, the z-score of height was calculated using the height of children and the corresponding median and standard deviation for children of the same age, which were obtained from the anthropometry guidelines.

We then further grouped nutritional status into four categories using the calculated z-score values. The first group consists of severely stunted children whose height-for-age z-scores are more than three standard deviations below the World Health Organization child growth standards median. The second group consists of stunted children whose height-for-age z-scores are more than two standard deviations below the WHO child growth standards median. The third group is considered to have normal nutritional status, and their z-score is within two standard deviations of the WHO child growth standards median. The last group consists of children who are considered to have above normal nutritional status, with z-scores more than two standard deviations above the WHO child growth standards median.

As the focus of our analysis in this research is stunted children, for our regression analysis, we categorized children into two groups—stunted and not stunted. The stunted group consisted of children who were severely stunted (as defined above) and was represented by a dummy variable equal to one. The remaining children were categorized as not stunted and were represented by a dummy variable equal to zero.

## 2.3. Individual-level variables

One of this study's particular aims was to assess the impact of children's dietary habits on stunting. The IFLS dataset provides data on dietary diversity and consumption frequency in the one week prior to the survey for each child in the family. Studies have suggested that adequate dietary diversity lowers the odds of stunting, whereas the consumption of unhealthy snacks raises it [14, 18–20]. We constructed a binary variable capturing the intensity of unhealthy snacking as a measure of children's dietary habits. Following Wang et al. [21], the dummy variable for unhealthy snacking took the value of 1 for children consuming unhealthy snacks more than 7 times a week (or more than once per day) and took the value of 0 for those with lower consumption frequency. Snacks in the IFLS survey include instant noodles, fast food, carbonated beverages and sweet snacks that are usually high in salt, fat and sugar and low in micronutrient content. However, unhealthy snack data are only available in IFLS wave 5. Therefore, we conducted two estimations. Our first estimation used both waves of the IFLS (wave 4 and wave 5) and excluded the unhealthy snack variable, whereas our second estimation used only the IFLS wave 5 and included the unhealthy snack variable.

Demographic characteristics of children, such as gender, have also been considered predictors of stunting [15, 22, 23]. A dummy variable was constructed to test whether male children have a higher risk of being stunted, as earlier studies have found [15, 22, 23]. The study also included neonatal weight as one of the predictors of stunting, as suggested by Tiwari et al. [23] and de Silva and Sumarto [15]. According to the World Health Organization [24], a newborn baby weighing fewer than 2.5 kg is considered a small baby. Our dummy variable assumed a value of 1 if the neonatal weight was fewer than 2.5 kg and 0 otherwise.

Infections (both clinical and subclinical), such as enteric infections and diarrhea, contribute to stunting [25]. The IFLS dataset has information on children's acute morbidity in the four weeks prior to the survey, including the occurrence of diarrhea (at least three times a day). We constructed a dummy variable that assumed a value of 1 for children who had experienced acute diarrhea (either with blood or mucous or who had a watery stool with a pale color) and 0 otherwise.

## 2.4. Household-level variables

Household-level variables consisted of both parental- and household-level factors that have been found to affect stunting in the wider literature. Previous studies have indicated that maternal education is associated with a lower risk of stunting [7, 15, 23]. Mothers are primary caregivers in most Indonesian households, and we controlled for maternal education using years of schooling. We also included the mother's height in our model to control for maternal stature, in line with previous literature [11, 15, 26]. According to Beal et al. [7], a short mother (fewer than 145 cm tall) has a higher risk of having stunted children. Therefore, we created a dummy variable equal to 1 for mothers whose height was fewer than 145 cm and 0 otherwise.

Previous studies have shown a strong association between stunting and socioeconomic background. De Silva and Sumarto [15] and Mani [27] presented evidence that higher household consumption expenditure per capita and greater wealth lower the prevalence of stunting among children. The IFLS dataset provides detailed information on food and nonfood consumption. Food expenditure consists of spending on staples, vegetables, dried foods, meat and fish, beverages and prepared food. The expenditure on nonfood items includes spending on education, electricity, water, telephone, household items, recreation, entertainment, clothing and medical costs. We used the quartile of total consumption expenditure per capita as a measure of household wealth, categorizing the expenditure data as quartiles labeled Q1 to Q4, with Q1 being the lowest quartile of expenditure. We included three dummy variables that identified poor households in the bottom quartile of the consumption expenditure distribution as a base (coded 0) compared with the higher quartiles of Q2, Q3 and Q4.

To account for possible differences in health service delivery between rural and urban areas, we included a dummy variable for place of residence, which equaled 1 for children living in rural areas and 0 for those living in urban areas. Some earlier studies have suggested that childhood stunting is more prevalent in rural areas [15, 23, 28], and we examined whether this also held true for Indonesia.

Finally, this study also examined the contribution of household access to WASH. Previous studies have underlined the importance of sanitation and access to health services in lowering the risk of stunting [12, 15, 29, 30]. This study considered the association between access to clean WASH and the risk of stunting. Three dummy variables were introduced in the empirical model to capture WASH variables. The main source of drinking water in households was coded as 1 if the source of drinking water was either tap water (piped water and groundwater) or mineral water and 0 otherwise. The availability of a toilet was coded as 1 if there was a toilet with its own septic tank for sanitation and 0 otherwise. Additionally, the disposal of garbage was coded as 1 if garbage was disposed into a trash can that is collected by a sanitation service and 0 otherwise.

## 2.5. Community-level variables

We examined the effect of nutrition-specific intervention programs in PHCs in young children's neighborhoods. The IFLS dataset provides information on access to three nutrition-specific services provided by PHCs: growth and development monitoring, additional treatment for malnutrition and additional nutrition for the poor. A dummy variable was created to differentiate between young children who have access to all three nutrition-specific services (coded as 1) and those who do not (coded as 0).

## 2.6. Statistical methodology

The study examined the individual-, household-, community- and province-level determinants of stunting using a multilevel mixed effects logistic model. Multilevel models allowed us

to take into account the hierarchical structure in our data and calculate residual components at each level in the hierarchy. The individual-level variables we controlled for included children's demographic characteristics, birthweight, history of diarrhea and dietary habits. The family-level variables included mother's education, mother's stature and family consumption. We also included contextual variables at the community and district levels in the form of access to clean water, hygiene and sanitation, availability of nutritional services and area of residence (rural/urban). The adjusted odds ratio (aOR) of the fixed effects reflects the likelihood of stunting in children. The intraclass correlation coefficient (ICC) measures the degree of homogeneity within clusters in the risk of stunting. The partitioning of the residual components at each level enables us to see the effect of "unobserved" province- and community-level characteristics on stunting. Another advantage of multilevel models is that they allow simultaneous estimation of group effects and the effect of group-level predictors.

We started our estimation by running mixed effects logit null models without covariates. The first null model introduced a random intercept term at level 4 (province level). The second null model included an additional random intercept term at level 3 (subdistrict level). Additionally, the third null model included an additional random intercept term at level 2 (household level). The null models provided an indication of how much variation in stunting each additional cluster accounts for. Predictors were introduced in a stepwise manner, starting with child-level covariates, and followed by family- or household- and community-level covariates.

## 3. Results

### 3.1. Characteristics of study participants

Table 1 depicts the children's characteristics in the two waves of the IFLS data. It can be seen from the table that the proportion of stunted children was 26.29%, which consisted of 8.88% severely stunted and 17.41% stunted children.

The table also shows that more than half of the children (56.43%) consumed unhealthy snacks frequently based on Wave 5 data. In terms of gender composition, 52% of the children were boys. Moreover, the proportion of children who suffered from acute diarrhea was 15.46%. However, only 4% had been small at birth, with a weight of less than 2.5 kg.

Mothers in our sample had, on average, 8 years of education, which is equivalent to reaching the second year of junior high school in Indonesia. In terms of stature, 45.17% were fewer than 145 cm tall and were, therefore, classified as short following Beal et al. [7]. For household-level variables, the data revealed that the monthly average household consumption expenditure per capita increases as we move from the bottom quartile to the top quartile. The average for the bottom quartile was 230,947 Rupiah (equivalent to $57.32 USD Purchasing Power Parity in 2014) compared to 2.3 million Rupiah (equivalent to $574.40 USD PPP in 2014) in the top quartile.

The proportion of children living in rural areas was 44.16%. From the descriptive statistics for access to WASH facilities, we can see that more than 96% of households had access to clean water, only 68.8% had access to sanitation, and an even lower proportion (approximately 32.8%) had access to hygiene. Last, the data show that 27% of families had access to all three nutrition-related services from PHCs.

### 3.2. Multilevel analyses of stunting determinants

We present the results from the estimation of our three null models in Table 2. As noted previously, the first null model includes random effects at the province level, the second null model includes random effects at both the province and the subdistrict levels, and the third null model includes random effects at the province, subdistrict and household levels.

**Table 1. Characteristics of study participants.**

|  | Summary statistics for 2007 & 2014 combined | | 2014 | | 2007 | |
|---|---|---|---|---|---|---|
| **Outcome variable** | | | | | | |
| Stunted (%) | 8045 | 26.29 (0.440) | 4,044 | 28.73 (0.45) | 4,001 | 23.82 (0.43) |
| **Independent variables** | | | | | | |
| *Individual (child-)level data* | | | | | | |
| Unhealthy snacking* | | | | | | |
| High frequency (%) | 2282 | 56.43 (0.496) | 2282 | 56.43 (0.496) | | |
| Low frequency (%)–base | 1762 | | 1762 | | | |
| Gender | | | | | | |
| Male (%) | 4183 | 52 (0.499) | 2116 | 52.32 (0.50) | 2067 | 51.66 (0.50) |
| Female—base | 3862 | | 1928 | | 1934 | |
| Baby size | | | | | | |
| Small baby (%) | 339 | 4.21 (0.200) | 176 | 4.35 (0.04) | 163 | 4.07 (0.04) |
| Normal weight baby—base | 7706 | | 3868 | | 3838 | |
| Diarrhea | | | | | | |
| Acute diarrhea (3 times/day in the past 4 weeks) (%) | 1244 | 15.46 (0.362) | 6959 | 17.19 (0.38) | 549 | 13.72 (0.34) |
| Not experienced acute diarrhea—base | 6801 | | 3349 | | 3452 | |
| *Household level data* | | | | | | |
| Mother's education (years of schooling)** | 8045 | 8.48 (4.272) | 4044 | 8.94 (4.23) | 4001 | 8.02 (4.27) |
| Mother's stature | | | | | | |
| Mother short (<145 cm) | 3634 | 45.17 (0.497) | 1979 | 48.94 (0.50) | 1655 | 41.36 (0.49) |
| Normal height (145 cm & above)–base | 4411 | | 2065 | | 2346 | |
| Consumption quartile (Rupiah) | | | | | | |
| First quartile—poor (Q1) | 2023 | 230,947 (107,719) | 1010 | 317,224 (82,955) | 1013 | 144,905 (38,446) |
| Second quartile (Q2) | 2018 | 413,914 (166,956) | 1010 | 570,282 (73,783) | 1008 | 257,236 (35,790) |
| Third quartile (Q3) | 2008 | 644,466 (243,638) | 1022 | 867,705 (1,077,70 | 986 | 413,076 (60,048) |
| Fourth quartile (Q4) | 1996 | 2,314,440 (4,812,522) | 1002 | 2,270,691 (2,227,326) | 994 | 2,358,541 (6,444,053) |
| Regional differences | | | | | | |
| Rural (%) | 3553 | 44.16 (0.497) | 1680 | 41.54 (0.49) | 1873 | 46.81 (0.50) |
| Urban (%)–base | 4492 | | 2364 | | 2128 | |
| Access to clean water (%) | 8045 | 96.35 (0.188) | 4044 | 96.98 (0.17) | 4001 | 95.70 (0.20) |
| Access to sanitation (%) | 8045 | 68.84 (0.463) | 4044 | 74.18 (0.44) | 4001 | 63.43 (0.48) |
| Access to hygiene (%) | 8045 | 32.78 (0.47) | 4044 | 36.35 (0.48) | 4001 | 29.17 (0.45) |
| *Community level variable* | | | | | | |
| Nutrition-specific intervention (%) | 7795 | 27 (0.444) | 3930 | 26.41 (0.44) | 3865 | 27.61 (0.45) |

*) Data on unhealthy snacking is only available in wave 5 of the IFLS.

**) Continuous variable of mother's years of schooling.

We can see from the table that the likelihood ratio test statistic is highly significant for all three null models (Models 1–3). This shows that multilevel models are a better fit to the data than a single-level model. Moreover, we also conducted likelihood ratio tests comparing the four-level model (with random effects at the province, subdistrict and household levels) with three- or two-level models and found that the four-level model was a better fit for the data. Thus, stunting in Indonesia varies by province, subdistrict and household, and analysis of stunting needs to take into account variations at all these levels.

**Table 2. Null models: With province effects (Model 1), province and subdistrict effects (Model 2) and province, subdistrict and household effects (Model 3).**

| Variable | Model 1 (95% CI) | Model 2 (95% CI) | Model 3 (95% CI) |
|---|---|---|---|
| Constant | -1.231 (-1.459, -1.004) | -1.371 (-1.572, -1.169) | -1.561 (-1.803, -1.318) |
| Between province variance | 0.186 (0.069, 0.503) | 0.116 (0.034, 0.378) | 0.144 (0.043, 0.472) |
| Between subdistrict variance | - | 0.378 (0.274, 0.522) | 0.452 (0.326, 0.648) |
| Between household variance | - | - | 0.795 (0.475, 1.329) |
| ICC (province) | 0.053 (0.0205, 0.133) | 0.0306 (0.009, 0.094) | 0.031 (0.009, 0.09) |
| ICC (province and subdistrict) | - | 0.131 (0.095, 0.177) | 0.127 (0.009, 0.174) |
| ICC (province, subdistrict and household) | - | - | 0.297 (0.226, 0.379) |
| Observations (young children) | 8045 | 8045 | 8405 |
| Group level | Province | Province; subdistrict | Province; subdistrict; household |
| Number of groups | 21 provinces | 21 provinces | 21 provinces |
| | | 1332 subdistricts | 1332 subdistricts |
| | | | 6437 households |
| Likelihood ratio test (LR) | 85.98 | 218.97 | 242.89 |
| Prob > chi2 | (0.000) | (0.000) | (0.000) |

Model 1: Two-level model (individual and household).

Model 2: Three-level model (individual, household, subdistrict).

Model 3: Four- level model (individual, household, subdistrict and province).

ICC: Intraclass correlation coefficient.

Fig 2 below shows a caterpillar plot of the residuals from all 1,332 subdistricts in the sample with 95 percent confidence intervals. The residual shows the departure of the risk of stunting for subdistricts from the average (overall mean). The graph reveals that for a substantial number of subdistricts, the 95% confidence interval does not overlap with the horizontal line at

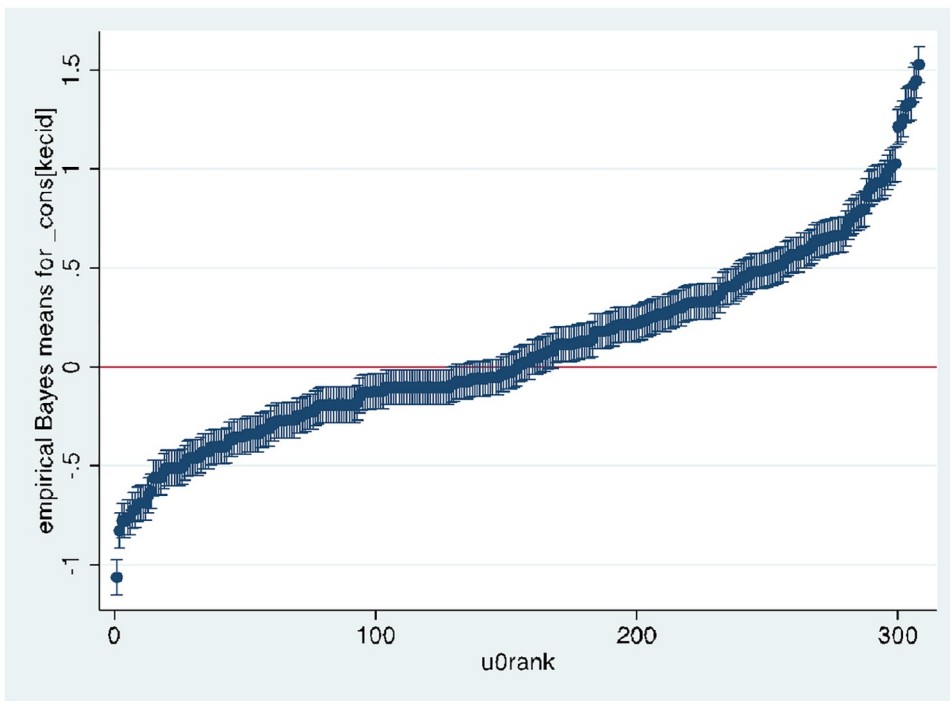

**Fig 2. Sub-districts caterpillar plot.**

zero, indicating that stunting in these subdistricts is significantly above average (above the zero line) or below average (below the zero line).

Table 2 also presents the intraclass correlation coefficient (ICC), which measures the degree of homogeneity in the risk of stunting within clusters (household/subdistrict/province). A higher ICC indicates a stronger correlation in the risk of stunting within a cluster. We can see from Model 3 that the level-4 intraclass correlation at the province level is 0.031, whereas the level-3 intraclass correlation at the subdistrict-within-province level is 0.127 and the level-2 intraclass correlation at the household-within-subdistrict level is 0.297. This indicates that within-subdistrict differences (i.e., differences between households and children) are more important in explaining the variation in the risk of stunting in Indonesia than differences between subdistricts and provinces.

We can also calculate the variance partition coefficients (VPCs) to see the contribution of unobserved cluster characteristics to the risk of stunting at each level in our model. The VPCs indicate that 3% of the variation in stunting is due to differences between provinces, whereas 10% is due to differences between subdistricts (within provinces). Differences between households (within-subdistricts) account for 17% of the variation, whereas individual differences between children account for 70% of the variation in stunting.

We present our estimation results in Table 3. Our most comprehensive model controls for household- and individual child-level covariates and community-level contextual characteristics (in particular, access to WASH). Two main estimates are generated from the stepwise regression in S1 Appendix, as discussed in the methodology section. The first estimation employs two complete waves of the dataset (waves 4 and 5); these results are presented in Columns 1 and 2 of Table 3. The second estimation relies on the latest wave of data (wave 5), as information about dietary habits, particularly unhealthy snacking behavior, is only available in wave 5; these results are presented in Table 3 Columns 3 and 4.

For the individual variables, our findings indicate that a high frequency of snack consumption is associated with a higher risk of children being stunted. In terms of magnitude, frequent snacking increases the risk of being stunted by 30% (95% CI 1.08–1.58), as shown in Table 3 (Column 3) [18, 28, 30–32, 34].

Additionally, our results indicate that boys have a higher risk of being stunted than girls, a finding that is consistent with previous studies. The likelihood of stunting was higher by 17% (95% CI 1.04–1.32) for boys than girls using two waves of data and by 26% (95% CI 1.04–1.51) using the Wave 5 dataset [32–35].

Another individual-level factor that is associated with a higher risk of stunting is neonatal weight. This study found that small babies weighing fewer than 2.5 kg at birth have a 2 times higher risk of being stunted than babies of normal weight. According to our model utilizing both waves of data (Table 3, Column 1), the odds of stunting for a small baby were 2.29 (95% CI 1.73–3.01).

The final individual-level predictor of stunting is infection with diarrhea. The estimations revealed that suffering from acute diarrhea (i.e., more than three times a day) in the four weeks prior to the survey is associated with a higher risk of being stunted, with an odds ratio of 1.27 (95% CI 1.08–1.49) in the model using Wave 4 data (Table 3, Column 3) and an odds ratio of 1.30 (95% CI 1.02–1.65) in the model using Waves 4 and 5 (Column 1).

Our results also indicate that maternal characteristics, such as maternal stature and years of schooling, and household characteristics, such as household wealth quartile, place of residence and access to basic infrastructure (WASH), are significantly associated with stunting. Children whose mothers are shorter than 145 cm have a 19% (95% CI [1.05–1.34] in the model using Waves 4 and 5) greater risk of being stunted. Moreover, the number of maternal years of

**Table 3. Multilevel mixed effects model logistic regression results of stunting status.**

| | Two waves of data (2007&2014) | | One wave of data (2014) | |
|---|---|---|---|---|
| Variables | Adjusted odds ratio (aOR) | (95% CI) | Adjusted odds ratio (aOR) | (95% CI) |
| **Individual level** | | | | |
| Unhealthy snacking | | | | |
| *High frequency* | - | - | 1.30*** | (1.08–1.58) |
| Gender | | | | |
| *Male* | 1.17*** | (1.04–1.32) | 1.26** | (1.04–1.51) |
| Baby size | | | | |
| *Small baby* | 2.29*** | (1.73–3.01) | 2.51*** | (1.63–3.88) |
| Diarrhea | | | | |
| *Acute diarrhea (3 times/day in the past 4 weeks)* | 1.27*** | (1.08–1.49) | 1.30** | (1.02–1.65) |
| **Family/household level** | | | | |
| Mother's education | 0.96*** | (0.94–0.97) | 0.94*** | (0.92–0.97) |
| Mother's stature | | | | |
| *Mother short (<145 cm)* | 1.19*** | (1.05–1.34) | 1.19* | (0.99–1.44) |
| Consumption quantile | | | | |
| *2nd quantile* | 0.77* | (0.65–0.91) | 0.76 | (0.59–0.98) |
| *3rd quantile* | 0.73*** | (0.61–0.87) | 0.71*** | (0.54–0.93) |
| *4th quantile* | 0.56*** | (0.46–0.68) | 0.50*** | (0.37–0.68) |
| **Community level** | | | | |
| Clean water | | | | |
| *No access* | 1.36* | (0.98–1.89) | 1.22 | (0.71–2.12) |
| Sanitation | | | | |
| *No access* | 1.27*** | (1.10–1.46) | 1.23* | (0.98–1.54) |
| Hygiene | | | | |
| *No access* | 1.52*** | (1.28–1.80) | 1.75*** | (1.34–2.27) |
| Regional differences | | | | |
| *Rural* | 1.19** | (1.02–1.40) | 1.10 | (0.86–1.41) |
| 2014 time dummy | 1.50*** | (1.32–1.70) | - | |
| *Intercept* | 0.18*** | (0.13–0.25) | 0.22*** | (0.13–0.37) |
| *Province effect—coef (std.dev)* | 0.13 (0.07) | | 0.20 (0.13) | |
| *Subdistrict effect—coef (std.dev)* | 0.19 (0.05) | | 0.44 (0.13) | |
| *Household effect—coef (std.dev)* | 0.79 (0.210) | | 1.31 (0.56) | |
| *Observations* | 8045 | | 4044 | |

Note: Unhealthy snacking data are only available in Wave 5 of the IFLS.

An odds ratio is statistically significant at either 1 percent (***), 5 percent (**) or 10 percent (*) of the confidence intervals.

schooling lowered the risk of being stunted (odds ratio of 0.96, 95% CI [0.94–0.97] in the model using data from Waves 4 and 5).

Our results also suggest that children from poor households have a higher risk of being stunted. The risk of stunting decreased by 23% (95% CI [0.65–0.91] in models using Waves 4 and 5 data [Table 3, Column 1]) for children in the second quartile of the wealth distribution. The risk of stunting was further lowered for children in the fourth quartile by 44% (95% CI [0.46–0.68] in the Waves 4 and 5 model).

It is apparent from our results that children living in rural areas have 19% (95% CI [1.02–1.40] in the Waves 4 and 5 model) greater odds of being stunted than children living in urban areas. This study also found that having no access to clean water increases the risk of being

stunted by 36% (odds ratio of 1.36, [95% CI (0.98–1.89)]). Similarly, having no access to sanitation was associated with a 27% higher risk of being stunted (odds ratio of 1.27, [95% CI (1.10–1.46)]). A higher risk of stunting was also associated with a lack of access to hygiene at 52% (odds ratio of 1.52, [95% CI (1.28–1.80)]).

The stepwise estimation using the multilevel mixed effects model shows that the risk of stunting is higher if the children have no access to nutrition services programs in PHCs, but this association is not statistically significant (odds ratio of 1.06, [95% CI (0.88–1.26)]). Finally, the estimation shows that the odds of being stunted for young children in 2014 was higher than that for young children in 2007 (odds ratio of 1.50, [95% CI (1.32–1.70)]). The data show that the proportions of stunted children in 2007 and 2014 were 24% and 29%, respectively.

## 4. Discussion

Our analysis found that stunting is associated with several individual-, family-/household- and community-level variables. Frequent unhealthy snacking, male sex, low birthweight and diarrhea increase the risk of being stunted. Family characteristics that contribute to stunting risk include short maternal stature and having a family with a low socioeconomic status. In terms of community characteristics, this study found that living in rural areas increases the risk of stunting by 19%. The risk of stunting is also higher for children living in a community with a lack of access to clean WASH.

The results show that there is a positive association between a high frequency of snack consumption and the risk of children being stunted. Studies have indicated that snacking is becoming more prevalent in Indonesia, both in rural and in urban areas. A study by Sekiyama et al. [19] showed that one-third of food consumed by young children in West Java, Indonesia can be categorized as snack food. High snack consumption has a detrimental effect on children's development because snacks contain mostly fat (59.6%) and energy (40%) but have a lower density of protein and micronutrients. Black et al. [31] and Tarwotjo et al. [32] suggested that a lack of micronutrient intake, such as calcium and vitamin A, adversely affects children's linear growth. The World Health Organization has reported that chronic deficiencies in micronutrients are experienced by more than 2 billion people worldwide [29]. Micronutrients are vital for children's development because they have a significant role in bone formation (calcium), long bone growth (zinc) and intrauterine femur length increase (supplements) [33–35].

The importance of micronutrients for children has also been established by other studies measuring the effect of the School Feeding Program (SFP) on children's development [36–38]. A quasi-experimental study by Metwally et al. [36] revealed that feeding children pie made of flour fortified with vitamins and minerals has a positive effect on cognitive development. Nevertheless, the effect of SFP on nutritional status takes a longer time to manifest, so the effect is not statistically significant. Another study of SFP in rural Kenya showed that the intake of minerals, such as zinc and iron, and vitamins may increase children's appetite, muscle growth and physical activity [37]. The importance of micronutritients on children's diet has also been found in the impact evaluation of SFP programs in rural Uganda [38]. It was found that children fed one or two eggs per day gained more height and weight because eggs are a source of 13 essential micronutrients and protein, which are essential for children's development.

Our finding that boys have a higher risk of being stunted than girls is consistent with previous studies. Adair and Guilkey [35], Moestue [34] and Wamani et al. [33], for example, found that the height-for-age z-score for girls is higher than that for boys; thus, girls have a lower prevalence of stunting in Sub-Saharan Africa and China. The literature suggests that higher stunting prevalence among boys may be explained by complementary feeding practices. Boys receive premature complementary foods, as parents perceive that breastfeeding is not

sufficient to fulfil the greater energy intake they believe is required for baby boys [18]. A study by Tumilowicz et al. [39] showed that among Guatemalan children, more boys than girls aged 2–3 months are fed complementary foods. The IFLS dataset for young Indonesian children also shows that complementary feeding is initiated for boys earlier than for girls. On average, complementary feeding for Indonesian young children in the IFLS data of Waves 4 and 5 started at 19.74 weeks for baby girls and 18.91 weeks for baby boys; the difference was statistically significant in a t-test.

In addition, these complementary feeding practices do not benefit boys because they are more likely to be fed more meals than girls. Tumilowicz et al. [39] presented evidence that baby boys are fed two more meals than girls in a 24-hour period. Premature complementary feeding practices have a detrimental effect on young children because they pose a greater risk of catching infectious diseases [40, 41]. In addition, early introduction of food before babies reach 6 months of age has no significant effect on children's length or weight development [23].

As presented in the Results section, we found that babies of lower birth weight (less than 2.5 kg at birth) have a two times higher risk of being stunted than babies of normal birthweight. This finding is consistent with a study by Tiwari et al. [23], which found that average and above average weight newborn babies have lower odds of being stunted than smaller babies. Furthermore, A Saleemi [42] and Varela-Silva et al. [43] also showed that the risk of smaller babies being stunted is three times higher than that for other babies.

According to Schmidt et al. [26], neonatal weight and particularly length are good indicators of the status of children's nutrition in the future. Low neonatal weight and short length may be an indication of intrauterine growth restriction (IUGR), meaning that babies are not growing at a normal rate inside the womb during pregnancy. Lower neonatal weight and height may also be related to maternal malnutrition during pregnancy, which in turn influences the development of the baby [23]. Moreover, small babies may also have been born prematurely, which means they may not have fully developed during pregnancy [23]. In the long run, IUGR may lead to numerous developmental issues, such as growth retardation, lower cognitive ability development and poor neurodevelopmental outcomes.

In terms of infection with diarrhea, our findings are consistent with previous studies by Bardosono et al. [44], Beal et al. [7] and Tiwari et al. [23]. These authors also found that diarrhea is associated with stunting. According to Richard et al. [45], diarrhea is a particularly common health issue in developing countries, as households lack access to clean water and sanitation. Drinking from unimproved water sources and drinking untreated water increase the risk of diarrhea due to intestinal infections from a variety of bacteria, parasites and viruses. Richard et al. [45] suggest that the impact of diarrhea on linear growth is particularly strong when young children suffer from multiple episodes of diarrhea in the first 24 months of their lives. Furthermore, diarrhea may lead to growth retardation if the occurrence of diarrhea coincides with a lack of good-quality food and poor access to health care. Diarrhea has a detrimental effect on linear and ponderal growth, as it lowers dietary intake, escalates metabolic demands and lessens nutrient absorption in the gut [45].

Our analysis found a strong association between maternal stature and childhood stunting. Semba et al. [11] similarly established a strong association between maternal stature and stunting, and Schmidt et al. [26] contended that maternal stature and neonatal weight are the strongest predictors of child stunting. Maternal stature is perceived to be a good indicator of intragenerational undernutrition. Prendergast and Humphrey [46] argued that having a stunted mother is relevant in explaining stunting prevalence in children because of the importance of the nutritional status of the mother on children's stunting. Mothers suffering from undernutrition may have a higher risk of having stunted children because of their significant influence, especially in the first 500 days of a child's life.

The association between maternal years of schooling and risk of stunting is also supported by previous studies [see, for instance, 8, 19, 41]. A better educated caregiver is perceived as having appropriate maternal nutritional knowledge [44]. Furthermore, Semba et al. [11] found that better educated parents tend to engage in more protective caregiving; for example, they may do this by ensuring that their children receive vitamin A capsules, are fully immunized, have access to better sanitation and consume iodized salt.

The results also provide evidence that children from poor households are at higher risk of being stunted. The strong association we found between family wealth and stunting has been established in the literature [7, 11, 13, 27, 47–50]. Bardosono, Sastroamidjoo and Lukito [44] argued that poor families lack the resources to consume high-quality nutritional foods and access health care.

Our result shows the positive association between living in rural areas and the risk of being stunted. Previous studies support the finding that children living in rural areas have a higher risk of stunting than their peers in urban areas [see, for instance, 15, 23, 28]. According to Mahendradata et al. [51] and Mulyanto, Kurst and Kringos [52], both the demand and the supply of health care vary between urban and rural areas. People living in urban areas have more access to health care and other related infrastructure, such as roads that reduce the travel time to health care facilities. Meanwhile, access to health services in rural areas is more limited. According to Sparrow and Vothknecht [53], 6.3% of subdistricts in Indonesia have no access to PHCs. These districts are mostly in rural areas outside Java Island. Furthermore, their study reported that 4.2% of PHCs in rural areas have no physician serving in health facilities. Physical health infrastructure, such as working incubators, lab facilities and outpatient polyclinics, is also more limited in rural areas.

Another study by Schmidt et al. [26] suggested that a higher prevalence of stunting in rural areas compared with urban areas is related to a greater sensitivity to changes in food prices. Families in rural areas are more sensitive to food price increases because they allocate two-fifths of their budget for staple needs. As the price of foods increases, the purchasing power of rural families declines, making it harder to fulfil the essential nutritional requirements of their children.

Our results also show that a lack of access to WASH is associated with stunting among young children in Indonesia. Secure access to WASH infrastructure is critical. Young children are more prone to diarrhea, intestinal worm infection and environmental enteropathy when households have poor WASH facilities [9]. These infections may lead to nutritional issues. For example, children may lose their appetites, so they might consume less food than they need. In addition, these types of infections can lead to the malabsorption of nutrition and chronic immune activation. Finally, infections may induce fever, which requires the body to burn more food and exert energy to fight the infection instead of using it for physical development.

The insignificant association we found between stunting and access to nutrition services in PHCs may be due to the limited capacity of the Indonesian nutrition program, as shown in a report by UNICEF [54]. In terms of coverage of the nutrition-specific interventions recommended by The Lancet [55], the Indonesian government has only adopted 4 out of 10. Of the remaining six programs, four are partially covered, and two are not included. In terms of health infrastructure, the proportion of nutritionists per head in the population is low in Indonesia and varies across regions. In the larger provinces on Java Island, such as in Central Java, the number of nutritionists to every 1,000 people was as high as 43.14 in 2017. However, the number was much lower in the provinces outside Java Island. For example, in East Nusa Tenggara, one of the provinces with a high prevalence of stunting, the number was only 12.04.

Our results also show that the risk of stunting for young children was higher in 2014 than in 2007. Previous studies have revealed that after a significant decline in stunting prevalence in

Indonesia in the 1990s and early 2000s [28], the prevalence remained unchanged in the 2000s [56]. Our finding aligns with national data from the Indonesia Socio-Economic Survey and Basic Health Survey, which shows that stunting prevalence was higher in 2013 than in 2007. A World Bank publication acknowledges the important role that nutrition programming and surveillance at the village level in the 1980s played in reducing the prevalence of malnutrition in Indonesia. However, this massive program has experienced setbacks and lost the close attention of the government. Furthermore, Indonesia has undergone decentralization, which has reduced the effectiveness of nutrition programs in improving children's nutritional status due to weak management and poor governance [57].

### 4.1. Study limitations and strengths

This study highlights the importance of family- and community-level variables in stunting, in addition to individual characteristics. The assessment of the different levels of clusters in this study (i.e., province, districts and households) facilitates understanding of how contextual factors contribute to stunting in children. Hence, our study demonstrates the urgency of addressing not only personal- or individual-level factors but also household- and community-level factors to reduce stunting prevalence.

The limitations of this study are listed here. First, regarding the source of the data, the two waves of the IFLS data used in our paper are representative of approximately 83% of the Indonesian population, covering 21 provinces; thus, some areas of eastern Indonesia were excluded from our analysis. Additionally, IFLS Wave 4 data do not capture unhealthy snacking behavior among young children. This variable is considered an important proxy for measuring children's dietary habits; therefore, we used only Wave 5 data to examine this issue. Second, from a methodological perspective, the cross-sectional nature of our analyses limited our ability to infer causation. Moreover, some data were based on self-reported information, for example, the birth weight of the children, and thus may be susceptible to measurement error. In addition, our study did not control for the random slope component in the model. The assessment of nutritional services was conducted from the supply side (i.e., the availability of nutritional services in the community). However, we adjusted for several important confounders and took into account some unobserved characteristics through multilevel modeling.

## 5. Conclusion

Stunting remains a development issue in Indonesia, with approximately 30% of young children being stunted. This study examined the multilevel determinants of stunting among young children in Indonesia. At the contextual level, the ICC showed that there is a correlation in the risk of stunting for children living in the same province. The correlation becomes stronger for children living in the same subdistricts. Finally, the strongest correlation of the risk of stunting was found among children living in the same household. Moreover, the likelihood ratio tests revealed that stunting in Indonesia varies by province, subdistrict and household level, and analysis of stunting needs to consider variations at all these levels.

At the child and family levels, our results identified several statistically significant determinants of childhood stunting. In terms of individual characteristics, being a boy, having a low neonatal weight and experiencing acute diarrhea were associated with stunting. In terms of family characteristics, we found that mothers' characteristics, specifically maternal stature and maternal education, were associated with stunting. In contrast, living in a family with a higher socioeconomic status lowered the risk of stunting, and the risk of stunting was much lower for children in the highest quartile of the wealth distribution. Finally, in terms of community characteristics, we found that living in rural areas increased the risk of stunting by 20%. The risk of

stunting was also higher for children living in a community with lack of access to clean WASH.

From a policy perspective, our findings suggest that tackling stunting in Indonesia requires substantial effort to create spaces that assist policy implementation in establishing supportive multilevel conditions. These include addressing both individual- and household-level factors that support good child nutrition and development. Healthy eating habits, mothers' education and awareness, socioeconomic characteristics and the availability of WASH matter.

## Supporting information

**S1 Appendix.**
(DOCX)

## Author Contributions

**Conceptualization:** Tri Mulyaningsih, Itismita Mohanty, Vitri Widyaningsih, Tesfaye Alemayehu Gebremedhin, Riyana Miranti, Vincent Hadi Wiyono.

**Data curation:** Tri Mulyaningsih, Vitri Widyaningsih.

**Formal analysis:** Tri Mulyaningsih.

**Funding acquisition:** Tri Mulyaningsih.

**Investigation:** Tri Mulyaningsih.

**Methodology:** Tri Mulyaningsih, Itismita Mohanty, Vitri Widyaningsih, Tesfaye Alemayehu Gebremedhin.

**Project administration:** Tri Mulyaningsih.

**Resources:** Tri Mulyaningsih.

**Supervision:** Itismita Mohanty, Tesfaye Alemayehu Gebremedhin, Riyana Miranti.

**Writing – original draft:** Tri Mulyaningsih.

**Writing – review & editing:** Itismita Mohanty, Vitri Widyaningsih, Tesfaye Alemayehu Gebremedhin, Riyana Miranti, Vincent Hadi Wiyono.

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
