## [Decision Letter · Decision Letter 0]

5 Jul 2021

PONE-D-21-06348

Beyond Personal Factor: Multilevel Determinants of Childhood Stunting in Indonesia

PLOS ONE

Dear Dr. Mulyaningsih,

Thank you for submitting your manuscript to PLOS ONE. After careful consideration, we feel that it has merit but does not fully meet PLOS ONE’s publication criteria as it currently stands. Therefore, we invite you to submit a revised version of the manuscript that addresses the points raised during the review process.

The study was focusing on assessing the Multilevel Determinants of Childhood Stunting in Indonesia. However, fundamental issues are indicated to be provided by aggressive editing for most of the study sections. Consider revising the spelling and grammar throughout the manuscript for increased clarity. 

Please note that your manuscript was reviewed by 2 experts in the field. There is consensus agreement that the idea of the article is interesting but also the majority detected sections that required additional work. The reviewers identified many important problems and provided copious comments (enclosed).

We look forward to receiving your revised manuscript.

Kind regards,

Ammal Mokhtar Metwally, Ph.D (MD)

Academic Editor

PLOS ONE

Journal Requirements:

Reviewers' comments:

Reviewer's Responses to Questions

**Comments to the Author**

1. Is the manuscript technically sound, and do the data support the conclusions?

Reviewer #1: Partly

Reviewer #2: Yes

2. Has the statistical analysis been performed appropriately and rigorously? 

Reviewer #1: Yes

Reviewer #2: Yes

3. Have the authors made all data underlying the findings in their manuscript fully available?

Reviewer #1: Yes

Reviewer #2: No

4. Is the manuscript presented in an intelligible fashion and written in standard English?

Reviewer #1: Yes

Reviewer #2: No

5. Review Comments to the Author

Reviewer #1: The statistical tests applied and the technicality of the manuscript are sound. However, the manuscript requires some revision with regard to its structure and language. The manuscript is very lengthy and difficult to read it.

Reviewer #2: MAJOR COMMENTS

(1)

How much missing data was there on each variable? How was missing data handled in the models?

(2)

The conclusion to this paper needs re-working. Too much of the conclusion is a repetition of material from earlier in the paper.

A suggested structure for the conclusion would be as follows:

• Brief review of main results

• Strengths of the study

• Limitations of the study, including any unobserved confounders which may affect the results

• Policy implications of the study

• Further work which should be undertaken using this data or in future studies

MINOR COMMENTS

There are a number of places in which the grammar could be improved.

TITLE OF PAPER

“Beyond Personal Factor: Multilevel Determinants of Childhood Stunting in Indonesia”

suggest

“Beyond Personal Factors: Multilevel Determinants of Childhood Stunting in Indonesia”

P 2

“Children health outcomes are poor in Indonesia despite the..”

suggest

“Child health outcomes are poor in Indonesia despite the..”

“Regarding to risk factor, Beal et al., (2018) (4), for example, establish…”

suggest

“With regards to risk factors, Beal et al., (2018) (4), for example, establish…”

P 3

“In term of protective factors of stunting, previous studies show…”

suggest

“In terms of protective factors for stunting, previous studies show…”

P 4

“…taking into account the clustering in the data generates a more reliable standard errors of regression coefficients…”

suggest

“…taking into account the clustering in the data generates more reliable standard errors of regression coefficients…”

“The survey has five waves of data: 1993, 1997, 2000, 2007-2008 and 2014-2015 rounds.”

suggest

“The survey has five waves of data, collected in 1993, 1997, 2000, 2007-2008 and 2014-2015.”

P 5

“The first wave of the survey had covered only 13 provinces but the number had been broadened to include…”

suggest

“The first wave of the survey covered only 13 provinces but the number has been broadened to include…”

“Using a-repeated cross-sectional survey…”

suggest

“Using a repeated cross-sectional survey…”

P 6

“The children health status data is a self-reported measure of general health status…”

suggest

“The child health status data is a self-reported measure of general health status…”

P 7

“Following Wang et al., (2018) (17), the dummy variable for unhealthy snacking will take on the value of 1 for children consuming unhealthy snacks for more than 7 times a week…”

suggest

“Following Wang et al., (2018) (17), the dummy variable for unhealthy snacking will take the value 1 for children consuming unhealthy snacks more than 7 times a week…”

“Demographic characteristics of children such as gender is also considered as a predictor…”

suggest

“Demographic characteristics of children such as gender are also considered as a predictor…”

P 8

“This study further constructs the expenditure into quartile data of the bottom 25 percentile (Q1), between 25-50 percentile (Q2), between 50-75 percentile (Q3) and those in the top 25 percentile (Q4).”

suggest

“This study encodes the expenditure data as quartiles, labelled Q1 to Q4, with Q1 being the lowest quartile of expenditure.”

P 12

“…incorporating mother stature, mother education and socio-economic status of the family…”

suggest

“…incorporating mother’s stature, mother’s education and socio-economic status of the family…”

“After cleaning the data, there are 8105 number of children used in the analysis.”

suggest

“After cleaning the data, there are 8105 children used in the analysis.”

P 18

“On average, complementary feeding is started at 19.74 weeks for baby girls and 18.91 weeks for baby boys and the difference is statistically significant using t-test.”

suggest

“On average, complementary feeding is started at 19.74 weeks for baby girls and 18.91 weeks for baby boys; this difference is statistically significant using a t-test.”

TITLE FOR TABLE 3

“Stepwise of Multilevel Mixed Effect Model Logistic Regression”

suggest

“Results of stepwise Multilevel Mixed Effect Logistic Regression Models”

P 22

“Children whose mothers are shorter than 145cm have 19 percent [95% CI (1.05-1.34) in model using wave 4 and 5] more risk to be stunted.”

suggest

“Children whose mothers are shorter than 145cm have a 19 percent [95% CI (1.05-1.34) in model using Waves 4 and 5] greater risk of being stunted.”

“Prendergast & Humphrey (2014) (39) argue that stunted mother is relevant in explaining stunting prevalence considering the importance of the nutritional status of the mother on children stunting.”

suggest

“Prendergast & Humphrey (2014) (39) argue that having a stunted mother is relevant in explaining stunting prevalence in children because of the importance of the nutritional status of the mother on child stunting.”

P 23

“Moreover, maternal years of schooling lowers the risk of being stunted [odds ratio of 0.96 95% CI (0.94-0.97) in model using two waves of 4 and 5].”

suggest

“Moreover, the number of maternal years of schooling lowers the risk of being stunted [odds ratio of 0.96 95% CI (0.94-0.97) in the model using data from Waves 4 and 5].”

“Our estimation results also suggest that children from poor households have a higher risk to be stunted.”

suggest

“Our results also suggest that children from poor households have a higher risk of being stunted.”

“According to Mahendradata et al (2017) (44) and Mulyanto, Kurst and Kringos (2019) (45), both demand and supply of health care are varied across urban and rural areas.”

suggest

“According to Mahendradata et al (2017) (44) and Mulyanto, Kurst and Kringos (2019) (45), both demand and supply of health care vary between urban and rural areas.”

P 24

“People living in the urban areas have more access to health care and other related infrastructure…”

suggest

“People living in urban areas have more access to health care and other related infrastructure…”

“The physical health infrastructure such as working incubators, lab facilities and outpatient polyclinics are more limited in the rural areas.”

suggest

“The physical health infrastructure such as working incubators, lab facilities and outpatient polyclinics are more limited in rural areas.”

“Families in rural areas are more sensitive to a food price increase because they allocate 2/5 of their budget for a staple.”

suggest

“Families in rural areas are more sensitive to food price increases because they allocate two fifths of their budget for staple needs.”

P 25

“In terms of health infrastructure, the proportion of nutritionists per population is low and varied across regions in Indonesia.”

suggest

“In terms of health infrastructure, the proportion of nutritionists per head of population is low and it varied across regions in Indonesia.”

“This finding is also correspondents with national data of Indonesia Socio-Economic Survey and Basic Health Survey that the stunting prevalence was remained high in 2013 compared to the 2007 survey and the figure is even slightly higher for 2013 survey.”

suggest

“This finding is in agreement with national data from the Indonesia Socio-Economic Survey and Basic Health Survey that stunting prevalence was higher in 2013 compared to the 2007 survey.”

P 26

“Nevertheless, the massive program had been set backs that government had loss attention and in the same time Indonesia undergo decentralization that reduced the effectiveness of nutrition program due to weak management and poor governance

(49).”

This sentence is unclear and needs re-writing.

6. PLOS authors have the option to publish the peer review history of their article (what does this mean?). If published, this will include your full peer review and any attached files.

Reviewer #1: **Yes: **Atnafu Mekonnen Tekleab

Reviewer #2: No

---

## [Author Response · Author response to Decision Letter 0]

18 Aug 2021

Responses to Reviewers

Article titled “Beyond Personal Factors: Multilevel Determinants of Childhood Stunting in Indonesia”

1. Is the manuscript technically sound, and do the data support the conclusions?

Reviewer #1: Partly

Reviewer #2: Yes 

Response: Thanks for your comment.

2. Has the statistical analysis been performed appropriately and rigorously?

Reviewer #1: Yes

Reviewer #2: Yes 

Response: Thanks for your comment.

3. Have the authors made all data underlying the findings in their manuscript fully available?

Reviewer #1: Yes

Reviewer #2: No 

Response: Thanks for your comment.

4. Is the manuscript presented in an intelligible fashion and written in standard English?

Reviewer #1: Yes

Reviewer #2: No 

Response: Thanks for your comment.

5. Review Comments to the Author

Reviewer #1: The statistical tests applied and the technicality of the manuscript are sound. However, the manuscript requires some revision with regard to its structure and language. The manuscript is very lengthy and difficult to read it.

Response: Thanks for your comment. We have streamlined the paper following your suggestion, for example we have shortened the method section. Regarding language, the paper has now been proofread by a professional editor. 

Reviewer #2: MAJOR COMMENTS

(1) How much missing data was there on each variable? How was missing data handled in the models?

Response: Thanks for pointing this out. We have revised the manuscript and provide a clearer description of the data in section 2, sub section 2.1. paragraph 2. 

“This study only covers young children aged five years old and below. The dataset includes a total of 8,290 young children, with 4,142 in wave 4 and 4,148 in wave 5. By using the child anthropometry information, we excluded children identified as having biologically implausible values (n=185, 2.2%) based on z-scores of height, following the World Health Organization (WHO) child growth standard reducing the number of children to 8105. In addition, we removed young children for whom complete information on household and community characteristics was not available (n=60, 0.7%). Hence, the number of young children used in our final analysis is 8,045. The proportion of respondents with missing values is insignificant, at 3 percent”.

(2) The conclusion to this paper needs re-working. Too much of the conclusion is a repetition of material from earlier in the paper.

A suggested structure for the conclusion would be as follows:

• Brief review of main results

• Strengths of the study

• Limitations of the study, including any unobserved confounders which may affect the results

• Policy implications of the study

• Further work which should be undertaken using this data or in future studies

Response: Thanks for your constructive feedback. We have restructured the conclusion accordingly to include the main results, the strengths of the study, the limitations of the study, policy implications and potential future research. 

Brief review of main results:

- The intraclass correlation coefficient (ICC) showed that there is a correlation of the risk of stunting for children living in the same province. The correlation becomes stronger for children living in the same sub-districts. Finally, the strongest correlation of the risk of stunting was found among children living in the same household. 

- Moreover, the likelihood ratio tests revealed that stunting in Indonesia varies by the province, sub-district and household level, and analysis of stunting needs to consider variations at all these levels. 

- This study conducted a stepwise estimation and consisted of two main estimations. The first estimation employed a complete set of data utilizing two waves of the survey and the second estimation relied on the wave 5 dataset. IFLS wave 4 data does not capture unhealthy snacking behavior among young children. This variable is considered an important proxy for measuring children’s dietary habits, and thus we used wave 5 data to examine this issue. 

- Our results show that frequent snacking increases the risk of being stunted. As this analysis shows that the associations between other covariates and stunting are robust, it is possible that if data had been available for the snacking variable in wave 4, the pattern of the association between child dietary habits and stunting might have also been found for that wave.

Strengths of the study

- Thus, this study adds value by taking into account the hierarchical structure of the data and the role of unobserved characteristics at the household, sub-district and provincial level. 

- This study also uses a repeated cross-sectional survey providing recent data about young children in Indonesia, which is an improvement on previous studies.

- This study focuses on the last two waves of the IFLS survey: wave 4 collected in 2007 and wave 5 collected in 2014. The data covers 8,045 young children aged five years old living in 21 provinces and 1,332 sub-districts in Indonesia, with 4,001 children in wave 4 and 4,044 children in wave 5. 

- Utilizing a multilevel mixed effects modelling takes the nesting in our data into account. Children are nested into households, which in turn are nested into sub-districts and with sub-districts further nested into provinces. Further, there are children living in the same sub-districts are nested into provinces. 

- In addition, the multilevel mixed effects model was appropriate for examining the effects of unobserved household, sub-district and provincial characteristics, as well as child level characteristics, on the probability of children being stunted. 

- This study highlights the importance of family and community level variables in stunting. 

- Our study also demonstrates the urgency of addressing not only personal or individual level factors, but also household and community level factors to reduce stunting prevalence. 

- The findings of this study are consistent with the results from previous studies. In terms of individual characteristics, being a boy, having a low neonatal weight and experiencing acute diarrhea are associated with stunting. 

- In terms of family characteristics, we found that mothers’ characteristics, specifically maternal stature and maternal education, are associated with stunting. In contrast, living in a family with better socio-economic status lowers the risk of stunting, and the risk of stunting is much lower for children in the highest quartile of the wealth distribution. 

- Finally, in terms of community characteristics, we found that living in rural areas increases the risk of stunting by 20 percent. The risk of stunting is also higher for children living in a community with lack of access to clean water, sanitation and hygiene.

Limitations

- Another data limitation associated with the IFLS sample is that the two waves of data used are representative of only about 83% of the Indonesian population, covering 21 provinces; thus, the analysis excludes some areas in eastern Indonesia. 

- There are other limitations of the study from a methodological perspective. First, the cross-sectional nature of our analyses limits our ability to infer causation. 

- Moreover, there were some potential biases or confounders in this study. Some data is based on self-reported information, for example the birth weight of the children, and thus may be susceptible to information bias. 

- Additionally, heterogeneity in the tastes and preferences of parents for health investment for their children was not able to controlled for. The assessment of nutritional services was conducted from the supply side (i.e., the availability of nutritional services in the community). However, we adjusted for several important confounders and have taken into account some unobserved characteristics through multilevel modelling. 

Policy implications

- From a policy perspective, our findings suggest that tackling stunting in Indonesia requires substantial effort to create spaces that assist policy implementation in establishing supportive multilevel conditions. These include addressing both individual and household-level factors that support good child nutrition and development. Healthy eating habits, mothers’ education and awareness, socio-economic characteristics and the availability of WASH matter. 

- In 2017, the Indonesian government established the National Strategy to Accelerate Stunting Prevention (StraNas Stunting), a four-year strategy to integrate critical services related to stunting across national, regional and community programs (54). The strategy involves 22 ministries covering the areas of health, early childhood education and development, water, sanitation and hygiene (WASH), food security, and social protection. It adopts a multi-sectoral approach. It aims to prevent 2 million children from becoming stunted between 2018 and 2022. However, our research relates to a period before the implementation of this national policy.

Future study:

- Thus, future research should incorporate this policy measure and aim to evaluate the impact of this strategy on decreasing stunting in Indonesia. 

- Future research may also expand the investigation of the determinants of stunting to eastern Indonesia, which is a less developed region than the western part of the country. The availability of the IFLS East survey conducted in 2012 will enable a similar analysis to be conducted in this region in the future.

MINOR COMMENTS

There are a number of places in which the grammar could be improved.

TITLE OF PAPER

“Beyond Personal Factor: Multilevel Determinants of Childhood Stunting in Indonesia”

suggest

“Beyond Personal Factors: Multilevel Determinants of Childhood Stunting in Indonesia”

P 2

“Children health outcomes are poor in Indonesia despite the..”

suggest

“Child health outcomes are poor in Indonesia despite the..”

“Regarding to risk factor, Beal et al., (2018) (4), for example, establish…”

suggest

“With regards to risk factors, Beal et al., (2018) (4), for example, establish…”

P 3

“In term of protective factors of stunting, previous studies show…”

suggest

“In terms of protective factors for stunting, previous studies show…”

P 4

“…taking into account the clustering in the data generates a more reliable standard errors of regression coefficients…”

suggest

“…taking into account the clustering in the data generates more reliable standard errors of regression coefficients…”

“The survey has five waves of data: 1993, 1997, 2000, 2007-2008 and 2014-2015 rounds.”

suggest

“The survey has five waves of data, collected in 1993, 1997, 2000, 2007-2008 and 2014-2015.”

P 5

“The first wave of the survey had covered only 13 provinces but the number had been broadened to include…”

suggest

“The first wave of the survey covered only 13 provinces but the number has been broadened to include…”

“Using a-repeated cross-sectional survey…”

suggest

“Using a repeated cross-sectional survey…”

P 6

“The children health status data is a self-reported measure of general health status…”

suggest

“The child health status data is a self-reported measure of general health status…”

P 7

“Following Wang et al., (2018) (17), the dummy variable for unhealthy snacking will take on the value of 1 for children consuming unhealthy snacks for more than 7 times a week…”

suggest

“Following Wang et al., (2018) (17), the dummy variable for unhealthy snacking will take the value 1 for children consuming unhealthy snacks more than 7 times a week…”

“Demographic characteristics of children such as gender is also considered as a predictor…”

suggest

“Demographic characteristics of children such as gender are also considered as a predictor…”

P 8

“This study further constructs the expenditure into quartile data of the bottom 25 percentile (Q1), between 25-50 percentile (Q2), between 50-75 percentile (Q3) and those in the top 25 percentile (Q4).”

suggest

“This study encodes the expenditure data as quartiles, labelled Q1 to Q4, with Q1 being the lowest quartile of expenditure.”

P 12

“…incorporating mother stature, mother education and socio-economic status of the family…”

suggest

“…incorporating mother’s stature, mother’s education and socio-economic status of the family…”

Response: Yes, we have revised accordingly.

p. 12

“After cleaning the data, there are 8105 number of children used in the analysis.”

suggest

“After cleaning the data, there are 8105 children used in the analysis.”

Response: Yes, thanks for your input. We have revised the sentence and provides more detail information regarding the data cleaning process in the note under table 3 as below:

• For model 1-4, we analyzed data from 8105 children, who met the inclusion criteria, and with anthropometric measures within the biologically plausible value.

• For model 5-10, we analyzed data from 8045 children, who met the inclusion criteria, and with anthropometric measures within the biologically plausible value, and with complete individual and household characteristics.

• For Model 11, we analyzed data from 7795 children who met the inclusion criteria, and with anthropometric measures within the biologically plausible value, and with complete individual and household characteristics, and also have information on nutritional services access. However, this model is not used in the following estimation, due to insignificant findings. 

P 18

“On average, complementary feeding is started at 19.74 weeks for baby girls and 18.91 weeks for baby boys and the difference is statistically significant using t-test.”

suggest

“On average, complementary feeding is started at 19.74 weeks for baby girls and 18.91 weeks for baby boys; this difference is statistically significant using a t-test.”

TITLE FOR TABLE 3

“Stepwise of Multilevel Mixed Effect Model Logistic Regression”

suggest

“Results of stepwise Multilevel Mixed Effect Logistic Regression Models”

P 22

“Children whose mothers are shorter than 145cm have 19 percent [95% CI (1.05-1.34) in model using wave 4 and 5] more risk to be stunted.”

suggest

“Children whose mothers are shorter than 145cm have a 19 percent [95% CI (1.05-1.34) in model using Waves 4 and 5] greater risk of being stunted.”

“Prendergast & Humphrey (2014) (39) argue that stunted mother is relevant in explaining stunting prevalence considering the importance of the nutritional status of the mother on children stunting.”

suggest

“Prendergast & Humphrey (2014) (39) argue that having a stunted mother is relevant in explaining stunting prevalence in children because of the importance of the nutritional status of the mother on child stunting.”

P 23

“Moreover, maternal years of schooling lowers the risk of being stunted [odds ratio of 0.96 95% CI (0.94-0.97) in model using two waves of 4 and 5].”

suggest

“Moreover, the number of maternal years of schooling lowers the risk of being stunted [odds ratio of 0.96 95% CI (0.94-0.97) in the model using data from Waves 4 and 5].”

“Our estimation results also suggest that children from poor households have a higher risk to be stunted.”

suggest

“Our results also suggest that children from poor households have a higher risk of being stunted.”

“According to Mahendradata et al (2017) (44) and Mulyanto, Kurst and Kringos (2019) (45), both demand and supply of health care are varied across urban and rural areas.”

suggest

“According to Mahendradata et al (2017) (44) and Mulyanto, Kurst and Kringos (2019) (45), both demand and supply of health care vary between urban and rural areas.”

P 24

“People living in the urban areas have more access to health care and other related infrastructure…”

suggest

“People living in urban areas have more access to health care and other related infrastructure…”

“The physical health infrastructure such as working incubators, lab facilities and outpatient polyclinics are more limited in the rural areas.”

suggest

“The physical health infrastructure such as working incubators, lab facilities and outpatient polyclinics are more limited in rural areas.”

“Families in rural areas are more sensitive to a food price increase because they allocate 2/5 of their budget for a staple.”

suggest

“Families in rural areas are more sensitive to food price increases because they allocate two fifths of their budget for staple needs.”

P 25

“In terms of health infrastructure, the proportion of nutritionists per population is low and varied across regions in Indonesia.”

suggest

“In terms of health infrastructure, the proportion of nutritionists per head of population is low and it varied across regions in Indonesia.”

“This finding is also correspondents with national data of Indonesia Socio-Economic Survey and Basic Health Survey that the stunting prevalence was remained high in 2013 compared to the 2007 survey and the figure is even slightly higher for 2013 survey.”

suggest

“This finding is in agreement with national data from the Indonesia Socio-Economic Survey and Basic Health Survey that stunting prevalence was higher in 2013 compared to the 2007 survey.”

Response: Yes we have revised accordingly.

P 26

“Nevertheless, the massive program had been set backs that government had loss attention and in the same time Indonesia undergo decentralization that reduced the effectiveness of nutrition program due to weak management and poor governance (49).”

This sentence is unclear and needs re-writing.

Response: Yes, we have revised it accordingly.

“However, this massive program has experienced setbacks and lost the close attention of government. Furthermore, at the same time Indonesia has undergone decentralization, which has reduced the effectiveness of nutrition programs in improving children’s nutritional status due to weak management and poor governance (53)”.

6. Abstract section: Not structured

Response: Thanks for your comment and we have revised the abstract accordingly, as follows:

Background: Stunting is still a major public health problem, including in Indonesia. Studies have reported the complexities of understanding the determinants associated with stunting. This study aims to examine the household, sub-district and province level determinants of stunting in Indonesia using a multilevel hierarchical mixed effects model. 

Methods: We used data from the Indonesian Family and Life Surveys (IFLS) waves 4 and 5 (for the years 2007 and 2014). Data from 8,045 children were analyzed. We included individual, family/household and community level variables in the analyses. A multilevel mixed effects model was employed to take into account the hierarchical structure of the data. Moreover, the model captures the effect of unobserved household, sub-district and province level characteristics on the probability of children being stunted. 

Results: Our findings showed that the odds of childhood stunting vary significantly by province, sub-district and household levels. Among the child-level covariates included in our model, dietary habits, neonatal weight, a history of infection, and gender significantly affect the risk of stunting. Household wealth status and parental education are significant household-level covariates associated with a higher risk of stunting. Finally, the risk of stunting is higher for children living in a community without access to water, sanitation and hygiene.

Conclusions: Stunting is associated with not only children’s individual factors, but also family and community level characteristics. Hence, interventions to reduce stunting should also take into account family and community characteristics to achieve effective outcomes. 

7. Introduction section: Authors mixed up reference citation styles (Vancouver vs Harvard styles) - see paragraph 1, lines 9-10. 

“Therefore, fighting against stunting still remains the main agenda for the government (Indonesia Medium Development Goals 2014-2019 and 2020-2024)”.

Response: Thanks for pointing this out and we have revised it accordingly.

“Therefore, fighting against stunting still remains the main agenda for the government as asserted in the Indonesia Medium Development Goals 2014-2019 and 2020-2024”.

8. Methods section:

• The authors have mentioned that there were 4,309 children in wave 4 and 4,235 children in wave 5 who were less than five years old. However, they also mentioned that there were 5,975 households who had children aged 5 and below. How can the number of households who had children under five be more than the number of children age less than five? Does that mean some of the children did belong to more than one households?

Response: Thanks for pointing this out. We have revised the draft in section 2, sub section 2.1. paragraph three and hopefully now it is clearer. The revision is below.

“Four hierarchical levels are considered in our analysis – individual (child), household, sub-district and province. Children (the lowest level in our mixed effects hierarchical model) are nested within households (level two). There are 6,437 households with 8,045 children aged 5 and below in the two waves of IFLS. Approximately 20 percent of the total households had more than one child aged 5 and below. Households are nested within sub-districts (Kecamatan in Indonesian). There are 1,332 sub-districts recorded in the two waves of the survey. 

• The authors need to elaborate the following issues:

o Which WHO curve was used to determine the nutritional status (stunting) of the children? The 2006 or….?

Response: Thanks for your question. This paper refers to the World Health Organization (2005) publication that is used by the Indonesian government in defining children’s nutritional status as stipulated by the Keputusan Menteri Kesehatan Republik Indonesia Nomor: 1995/Menkes/SK/XII/2010 .

o Who determined the nutritional status (stunting), the data collectors at the time of the data collection or the authors of this manuscript (by analyzing the raw data i.e. height-to-age of the child)?

Response: Thanks for your question. We (authors) determined the nutritional status by analyzing the raw data (i.e. height-to-age of the child).

• The community level variable that is “access to nutrition specific services” lacks clarity. The authors categorized the variable based on the accessibility of a child to three nutrition specific services provided by PHC in the area. What does “access” means? Does it necessarily imply the utilization of the mentioned services? How can a child be categorized if those mentioned services are available but child is not utilizing any of the services?

Response: Thanks for pointing this out. The access to nutritional specific services refers to the supply of the services in the neighborhood. Thus, we added this as a limitation of the study in the conclusion, as follows: 

“Some data is based on self-reported information, for example the birth weight of the children, and thus may be susceptible to information bias. Additionally, heterogeneity in the tastes and preferences of parents for health investment for their children was not able to controlled for. The assessment of nutritional services was conducted from the supply side (i.e., the availability of nutritional services in the community). However, we adjusted for several important confounders and have taken into account some unobserved characteristics through multilevel modelling”. 

• I am not sure if it is necessary to describe the details of the mathematical formulas that were used by the authors to run the different prediction models as it is done by the authors of this manuscript. Because of such lengthy description, it is no attractive for reading. Better only to state the models and why those models are preferred. 

Response: Thanks for your comment. We have streamlined the description of the methodology and kept the final empirical model of equation 6. 

“The final model in which we estimate determinants of stunting, adjusting for provincial, district, community and family unobserved characteristics, is presented in the equation below: 

logit (y_iljk=1| x_ijk,ω_ljk,u_jk,v_k )= β_0+ β_1 X_(1iljk )+ β_2 X_2ljk+ β_3 X_3jk+ β_4 X_4k+ v_0k+u_0jk+ω_0ljk

Where β_1 is the vector of coefficients for predictor variables at level 1 (X_(1iljk )); β_2 is the vector of coefficients for predictor variables at level 2 (X_2ljk); β_3 is the vector of coefficients for predictor variables at level 3 (X_3jk); and β_4 is the vector of coefficients for predictor variables at level 4 (X_4k).

• Section 2.7 describes the authors’ findings. But it is kept under the “Data and Methods” section. Better to move it to the “Result section” as section 2.7 contains the research findings not research method.

Response: Thanks for pointing this out. We have relocated section 2.7 to the results section.

9. Section 2.7. characteristics of study participants:

• Table 1: Standard deviation the variable “stunted” is mentioned as 0.440. How can a categorical variable have a standard deviation?

• Table 1 is confusing and difficult to understand because of the following reasons:

o The purpose of calculating “standard deviation” for categorical variables is not clear.

o Look at the following two observations which are directly taken from the table ( similar problem exists for other variables too):

Gender

Male= 8105

Female= 0

Baby size

Small baby= 8105

Normal baby=0

From the above observations, what readers can understand is that there were no female babies and normal size babies.

Response: Table 1 has been revised accordingly and we hope now this is clearer.

• Table 4: The table cells contain “starred” values. What do those “stars” imply? Needs description as foot note under the table.

Response: Thanks for your question. We have added information below table 4 as below:

odds ratio is statistically significant at either 1 percent (***), 5 percent (**) or 10 percent (*) of the confidence intervals.

• In the methods section the authors mentioned that there were 4,309 children in wave 4 and 4,235 in wave 5 of the survey. However, in the result section they mentioned that they have included 4015 children of wave 4 and 4090 children of wave 5 (Total of 8105 children). The authors need to explain how and why some of the children were excluded from the analysis.

Response: Thanks for pointing this out. We have revised the manuscript and provide a clearer description of the data in section 2, sub section 2.1. paragraph 2. 

“This study only covers young children aged five years old and below. The dataset includes a total of 8,290 young children, with 4,142 in wave 4 and 4,148 in wave 5. By using the child anthropometry information, we excluded children identified as having biologically implausible values (n=185, 2.2%) based on z-scores of height, following the World Health Organization (WHO) child growth standard reducing the number of children to 8105. In addition, we removed young children for whom complete information on household and community characteristics was not available (n=60, 0.7%). Hence, the number of young children used in our final analysis is 8,045. The proportion of respondents with missing values is insignificant, at 3 percent”.

• The total number of children included in the analysis is 8105 (the sum of children in wave 4 and wave 5). It is not clear why the authors summed up the number of children from the two waves and treated the two group of populations as single population. The population of children in wave 4 and wave 5 are separate since data collected from the two group of population was collected at different times (2007 and 2014) and should not be summed up and should not be treated as homogenous population. 

Response: Thanks for the feedback. Yes, we have added characteristics of study participants in table 1 for the two waves separately (2014 and 2007) and the combined two waves. 

• Result and discussion are mixed. I advise the authors to follow the journal’s template.

Response:

Thanks for your comment. Yes, we have divided results and discussion into separate sections.

• Authors need to mention the limitation of their study

Response: Thanks for your comment. Yes, we have added the study’s limitations in the conclusion, as follows. 

- Another data limitation associated with the IFLS sample is that the two waves of data used are representative of only about 83% of the Indonesian population, covering 21 provinces; thus, the analysis excludes some areas in eastern Indonesia. 

- There are other limitations of the study from a methodological perspective. First, the cross-sectional nature of our analyses limits our ability to infer causation. 

- Moreover, there were some potential biases or confounders in this study. Some data is based on self-reported information, for example the birth weight of the children, and thus may be susceptible to information bias. 

- Additionally, heterogeneity in the tastes and preferences of parents for health investment for their children was not able to controlled for. The assessment of nutritional services was conducted from the supply side (i.e., the availability of nutritional services in the community). However, we adjusted for several important confounders and have taken into account some unobserved characteristics through multilevel modelling. 

10. Conclusion:

• Under this section, the authors tried to summarize their manuscript rather than putting the main finding of their research. Better to revise this section in such a way.

Response: Thanks for your comment. The conclusion has been revised accordingly based on the below structure for the conclusion as follows:

• Brief review of main results

• Strengths of the study

• Limitations of the study, including any unobserved confounders which may affect the results

• Policy implications of the study

• Further work which should be undertaken using this data or in future studies.

---

## [Decision Letter · Decision Letter 1]

20 Sep 2021

PONE-D-21-06348R1Beyond Personal Factor: Multilevel Determinants of Childhood Stunting in IndonesiaPLOS ONE

Dear Dr. Mulyaningsih,

Thank you for submitting your manuscript to PLOS ONE. After careful consideration, we feel that it has merit but does not fully meet PLOS ONE’s publication criteria as it currently stands. Therefore, we invite you to submit a revised version of the manuscript that addresses the points raised during the review process.

Great effort was made by the authors to utilize the feedback that was provided for them to correct. I find it interesting and improved with respect to the original submission. However, there are still major things to adjust in addition to the enclosed reviewers’ comments.  Despite the known influence of the dietary behaviours and habits on growth and the effect of the national school feeding programs on improving stunting among school children, yet the introduction and discussion lack a lot of references concerning national figures for stunting and its determinants among children in low- and middle-income countries with similar context. Commenting on the impact of the national school Feeding programs on growth and subsequently on development and school achievement as a solution and remedy way for improving stunting in the discussion section (which is vital) is still missing. Please elaborate on this in structural relationships within the discussion, and recommendations.

Please consider reviewers’ comments for more details  

We look forward to receiving your revised manuscript.

Kind regards,

Ammal Mokhtar Metwally, Ph.D (MD)

Academic Editor

PLOS ONE

Reviewers' comments:

Reviewer's Responses to Questions

**Comments to the Author**

1. If the authors have adequately addressed your comments raised in a previous round of review and you feel that this manuscript is now acceptable for publication, you may indicate that here to bypass the “Comments to the Author” section, enter your conflict of interest statement in the “Confidential to Editor” section, and submit your "Accept" recommendation.

Reviewer #3: (No Response)

Reviewer #4: (No Response)

2. Is the manuscript technically sound, and do the data support the conclusions?

Reviewer #3: Yes

Reviewer #4: Partly

3. Has the statistical analysis been performed appropriately and rigorously? 

Reviewer #3: Yes

Reviewer #4: Yes

4. Have the authors made all data underlying the findings in their manuscript fully available?

Reviewer #3: Yes

Reviewer #4: Yes

5. Is the manuscript presented in an intelligible fashion and written in standard English?

Reviewer #3: Yes

Reviewer #4: No

6. Review Comments to the Author

Reviewer #3: All comments are addressed properly. The manuscript is technically sound, statistical analysis has been performed appropriately and rigorously and manuscript is presented in an intelligible fashion but need some grammatical corrections.

Reviewer #4: 1. The manuscript is unnecessarily long such that the core issues are masked

2. The methodology section need not include detailed formulae used in deriving the results. This section needs revising to make it shorter and succinct

3. The results tables are too crowded. Summarised tables can be given in the results and more elaborate tables provided as addendum, if necessary

4. The conclusions read like another discussion of results. This section needs to be focussed on answering the objectives of the study

7. PLOS authors have the option to publish the peer review history of their article (what does this mean?). If published, this will include your full peer review and any attached files.

Reviewer #3: No

Reviewer #4: No

---

## [Author Response · Author response to Decision Letter 1]

19 Oct 2021

Response to specific reviewer and editor comments.

1. If the authors have adequately addressed your comments raised in a previous round of review and you feel that this manuscript is now acceptable for publication, you may indicate that here to bypass the “Comments to the Author” section, enter your conflict of interest statement in the “Confidential to Editor” section, and submit your "Accept" recommendation.

Reviewer #3: (No Response)

Reviewer #4: (No Response)

Response: We have revised the manuscript according to editor and reviewers’ inputs as described below.

2. Is the manuscript technically sound, and do the data support the conclusions?

Reviewer #3: Yes

Reviewer #4: Partly

Response: Thanks for your comment. We have revised the conclusion as per reviewer #4 suggestion in point 6 below.

3. Has the statistical analysis been performed appropriately and rigorously?

Reviewer #3: Yes

Reviewer #4: Yes

Response: Thanks for your comment.

4. Have the authors made all data underlying the findings in their manuscript fully available?

Reviewer #3: Yes

Reviewer #4: Yes

Response: Thanks for your comment.

5. Is the manuscript presented in an intelligible fashion and written in standard English?

Reviewer #3: Yes

Reviewer #4: No

Response: The manuscript has been proof read by American Journal Experts to assure that the manuscript is presented in an intelligible fashion, clear, correct, unambiguous and have no grammatical errors. 

6. Review Comments to the Author

Reviewer #3: All comments are addressed properly. The manuscript is technically sound, statistical analysis has been performed appropriately and rigorously and manuscript is presented in an intelligible fashion but need some grammatical corrections.

Response: The manuscript has been proof read by American Journal Experts to assure that the manuscript is presented in an intelligible fashion, clear, correct, unambiguous and have no grammatical errors. 

Reviewer #4: 

1. The manuscript is unnecessarily long such that the core issues are masked

2. The methodology section need not include detailed formulae used in deriving the results. This section needs revising to make it shorter and succinct

Response: Regarding reviewer#4 input number 1 and 2, we have removed the empirical model in the methodology section and make the section shorter. Hopefully, the revised manuscript is more concise. 

3. The results tables are too crowded. Summarised tables can be given in the results and more elaborate tables provided as addendum, if necessary

Response: Thanks for the suggestion. We relocated table 3 of stepwise results into appendix.

4. The conclusions read like another discussion of results. This section needs to be focussed on answering the objectives of the study.

Response: Thanks for the suggestion. We have revised the conclusion so it is more focus on answering the objectives of the study. Moreover, we relocated the study limitation from conclusion into discussion. We added a subsection at the end of the discussion section to explain the study limitation.

7. PLOS authors have the option to publish the peer review history of their article (what does this mean?). If published, this will include your full peer review and any attached files.

Do you want your identity to be public for this peer review? For information about this choice, including consent withdrawal, please see our Privacy Policy.

Reviewer #3: No

Reviewer #4: No

Response: Yes, noted.

8. Despite the known influence of the dietary behaviours and habits on growth and the effect of the national school feeding programs on improving stunting among school children, yet the introduction and discussion lack a lot of references concerning national figures for stunting and its determinants among children in low- and middle-income countries with similar context. Commenting on the impact of the national school Feeding programs on growth and subsequently on development and school achievement as a solution and remedy way for improving stunting in the discussion section (which is vital) is still missing. Please elaborate on this in structural relationships within the discussion, and recommendations.

Response: 

Thanks for the suggestion. 

- We have added a paragraph in the introduction presenting data concerning figures for stunting among children in low- and middle-income countries with similar context. 

- We have added citation of articles discussing the effect of the national school feeding programs on improving nutritional status among school children in the discussion section to explain the role of dietary habits on children nutritional status. 

- There are three additional literatures that we have added to explain the importance of dietary habits in lowering risk of nutritional issues such as stunting as below.

1. Metwally AM, El-Sonbaty MM, el Etreby LA, Salah El-Din EM, Abdel Hamid N, Hussien HA, et al. Impact of National Egyptian school feeding program on growth, development, and school achievement of school children. World Journal of Pediatrics. 2020 Aug 13;16(4).

2. Murphy SP, Gewa C, Liang L-J, Grillenberger M, Bwibo NO, Neumann CG. School Snacks Containing Animal Source Foods Improve Dietary Quality for Children in Rural Kenya. The Journal of Nutrition. 2003 Nov 1;133(11).

3. Baum JI, Miller JD, Gaines BL. The effect of egg supplementation on growth parameters in children participating in a school feeding program in rural Uganda: a pilot study. Food & Nutrition Research. 2017 Jan 6;61(1).

---

## [Decision Letter · Decision Letter 2]

8 Nov 2021

Beyond Personal Factor: Multilevel Determinants of Childhood Stunting in Indonesia

PONE-D-21-06348R2

Dear Dr. Mulyaningsih,

We’re pleased to inform you that your manuscript has been judged scientifically suitable for publication and will be formally accepted for publication once it meets all outstanding technical requirements.

Kind regards,

Ammal Mokhtar Metwally, Ph.D (MD)

Academic Editor

PLOS ONE

Additional Editor Comments (optional):

A great effort was made by the authors to utilize the feedback that was provided for them to correct. I find it interesting and improved with respect to the original submission.

Reviewers' comments:

Reviewer's Responses to Questions

**Comments to the Author**

1. If the authors have adequately addressed your comments raised in a previous round of review and you feel that this manuscript is now acceptable for publication, you may indicate that here to bypass the “Comments to the Author” section, enter your conflict of interest statement in the “Confidential to Editor” section, and submit your "Accept" recommendation.

Reviewer #3: All comments have been addressed

Reviewer #4: All comments have been addressed

2. Is the manuscript technically sound, and do the data support the conclusions?

Reviewer #3: Yes

Reviewer #4: Yes

3. Has the statistical analysis been performed appropriately and rigorously? 

Reviewer #3: Yes

Reviewer #4: Yes

4. Have the authors made all data underlying the findings in their manuscript fully available?

Reviewer #3: Yes

Reviewer #4: Yes

5. Is the manuscript presented in an intelligible fashion and written in standard English?

Reviewer #3: Yes

Reviewer #4: Yes

6. Review Comments to the Author

Reviewer #3: The manuscript is written in an intelligible fashion and statistical analysis are performed appropriately and rigorously.

Reviewer #4: (No Response)

7. PLOS authors have the option to publish the peer review history of their article (what does this mean?). If published, this will include your full peer review and any attached files.

Reviewer #3: No

Reviewer #4: No

---

## [Editor Report · Acceptance letter]

11 Nov 2021

PONE-D-21-06348R2 

Beyond Personal Factors: Multilevel Determinants of Childhood Stunting in Indonesia 

Dear Dr. Mulyaningsih:

I'm pleased to inform you that your manuscript has been deemed suitable for publication in PLOS ONE. Congratulations! Your manuscript is now with our production department. 

Kind regards, 

on behalf of

Professor Ammal Mokhtar Metwally 

Academic Editor

PLOS ONE